# Hydrophilic Sulfonate Covalent Organic Frameworks for Serum Glycopeptide Profiling

**DOI:** 10.3390/ijms26051957

**Published:** 2025-02-24

**Authors:** Shishu Yang, Yuheng Jiang, Shijie Jiang, Lihong Liu, Si Liu, Hua Zhang, Zhiyuan Gu

**Affiliations:** 1Key Laboratory of Green Chemical Media and Reactions, Ministry of Education, Henan Key Laboratory of Organic Functional Molecule and Drug Innovation, School of Chemistry and Chemical Engineering, Henan Normal University, Xinxiang 453007, China; zj2791567239@163.com (Y.J.); jiangsj5555@163.com (S.J.); liulh@htu.edu.cn (L.L.); 2Department of Epidemiology and Health Statistics, School of Public Health, Fujian Medical University, Fuzhou 350122, China; siliu@fjmu.edu.cn; 3Jiangsu Key Laboratory of Biofunctional Materials, Jiangsu Collaborative Innovation Center of Biomedical Functional Materials, College of Chemistry and Materials Science, Nanjing Normal University, Nanjing 210023, China; guzhiyuan@njnu.edu.cn

**Keywords:** covalent organic frameworks, glycopeptides, enrichment, serum, mass spectrometry

## Abstract

Aberrant protein glycosylation is closely associated with a number of biological processes and diseases. However, characterizing the types of post-translational modifications (PTMs) from the complex biological samples is challenging for comprehensive glycoproteomic analysis. The development of high-performance enrichment materials and strategies during the sample pretreatment process is a prerequisite to glycoproteome research. Here in this work, a sulfonate-rich covalent organic framework (COF) called TpPa-(SO_3_H)_2_ (referred to as SCOF-2) was synthesized using the Schiff base reaction for the identification of glycopeptides. Benefiting from high hydrophilicity and abundant sulfonate affinity, a total of 28 and 16 glycopeptides could be efficiently detected from the standard glycoproteins of horseradish peroxidase (HRP) and immunoglobulin G (IgG) tryptic digest, respectively. Moreover, the as-prepared sulfonate-rich SCOF-2 has an ultralow detection limit (0.01 fmol μL^−1^), excellent enrichment selectivity (molar ratio HRP:BSA = 1:5000), satisfactory recovery rate (89.1%), high adsorption capacity (150 mg g^−1^) and good reusability in the individual enrichment. Meanwhile, by using the SCOF-2 adsorbent, 196 and 194 endogenous glycopeptides in the serum of ovarian cancer patients and healthy people among triplicates were successfully enriched and identified, respectively, using combined nanoLC–MS/MS technology. It demonstrated its great application potential in glycoproteomics research and provided a novel insight for the design of affinity materials.

## 1. Introduction

Post-translational modifications (PTMs), which are chemical covalent modifications and most often dynamically regulated by enzymes [1,2], play significant and prevalent roles in numerous biological processes and behaviors of life [3,4]. Protein glycosylation, one of the most ubiquitous and representative PTMs, exerts a lot of impacts on protein stability, folding, distribution, and activity [5,6]. Meanwhile, aberrant glycosylation has been reported to be related to the occurrence and development of various diseases, including cancer, Alzheimer’s disease, cardiovascular disease and so on [7,8,9]. In other words, the traits of protein glycosylation can provide primary information for clinical diagnosis and the progression of disease [10]. Consequently, the elucidation of glycosylated proteins is essential for understanding disease mechanisms [11]. Mass spectrometry (MS) technology is regarded as a powerful tool for the comprehensive profiling of glycoproteomes due to its excellent sensitivity and high throughput ability [12,13,14]. However, it is challenging to directly detect glycoproteins and glycopeptides from complicated biological samples owing to their extremely low abundance, poor ionization efficiency, the interference of co-existing non-glycopeptides and the inherent heterogeneity of glycans [15]. Therefore, the effective enrichment of trace glycopeptides from highly complex mixtures prior to MS analysis is of great importance for successful glycopeptide identification.

To date, a variety of enrichment methods have been constructed as efficient strategies for the capture of glycopeptides by disparate enrichment mechanisms, including boric acid affinity chromatography (BAAC) [16,17], hydrazide chemistry [18,19,20], lectin affinity chromatography [21,22], and hydrophilic interaction liquid chromatography (HILIC) [23,24,25]. Among them, the principle of the HILIC method, mainly relying on the differences between hydrophilic glycopeptides and hydrophobic non-glycopeptides, could achieve unbiased enrichment ability towards different glycopeptides with high sensitivity and excellent reproducibility. Currently, plenty of HILIC-based materials, such as hydrophilic monolithic columns [26,27], modified magnetic nanoparticles [28,29], modified silica materials [14,30], polymers [31,32] and metal–organic frameworks [33,34] have been widely reported. Unfortunately, due to the limited glycopeptide-specific recognition sites, unsuitable mass transfer kinetics and low relative density of hydrophilic groups, traditional HILIC materials usually suffer from low glycopeptide binding selectivity and low detection sensitivity. Thus, the construction of affinity materials with outstanding hydrophilicity is still a highly desirable work to realize efficient glycopeptide enrichment and in-depth profiling of glycoproteomic research.

Covalent organic frameworks (COFs) stand for an emerging category of porous crystalline materials, which are formed by robust covalent bonds of organic building blocks based on dynamic covalent chemistry (DCC) that are mainly composed of H, C, N, O, and B elements [35,36,37]. In comparison with other types of porous materials, COFs possess the merits of permanent porosity, tunable pore size, relatively high thermal and chemical stability, large surface area and low crystal density [38,39], which establishes them as ideal candidates for wide applications in different fields including gas storage [40], separation [41,42], optoelectricity [43], sensing [44], catalysis [45], etc. Recently, taking advantage of the splendid performance of COFs, their applications in proteomics to enrich targets from complex biological samples have attracted more attention. For example, Zhou’s group successfully synthesized a hydrophilic amino-functionalized TpPa-1 COF that showed good selectivity for glycopeptides [46]. Zhang’s group introduced hydrophilic glutathione to improve the hydrophilicity of COFs (carboxyl-functionalized) using a post-synthetic modification method and enhanced the enrichment performance toward glycopeptides [47]. According to previously reported works, we noticed that amino, carboxyl or hydroxyl-functionalized COFs were generally selected as hydrophilic adsorbents to enrich glycopeptides. However, the capture sensitivity and selectivity of these materials were not so satisfactory and it was complicated and tedious to design the route of post-synthetic modification. As a consequence, developing functionalized COFs with more hydrophilic groups and more glycopeptide recognition sites to improve enrichment performance is among the mainstream pursuits.

Sulfonyl groups exhibit excellent hydrophilicity and chemical stability; however, sulfonyl-functionalized materials for the selective enrichment of glycopeptides have rarely been reported [48,49,50]. Therefore, it was first introduced here in this work to functionalize the as-prepared COFs. Here in this work, the rational selection from a range of sulfonate-rich COFs and a non-sulfonate TpPa COF are shown. The TpPa-(SO_3_H)_2_ (referred to as SCOF-2) together with another three, TpPa-SO_3_H, TpBD-(SO_3_H)_2_, and TFPB-BD-(SO_3_H)_2_ (denoted as SCOF-1, SCOF-3, and SCOF-4), were obtained for the successful enrichment of glycopeptides. SCOF-2 demonstrated excellent selectivity and the results of the theoretical calculations were consistent with the Experimental Section (Figure 1). Moreover, it showed extremely low detection limits as well as reusability and binding capacity in glycopeptide enrichment. Thereafter, six human serum samples (three healthy volunteers and three patients with ovarian cancer) were enriched using SCOF-2 to evaluate the performance of the sulfonate-rich COFs in the pretreatment of real biological samples. The implementation of this work will offer a new approach for the effective separation and identification of glycoproteins, and further establish meaningful work for the application of COF materials in post-translational modified proteomics.

## 2. Results

### 2.1. Characterization of the Sulfonate-Rich COFs and TpPa COF

SCOF-2 was synthesized based on a conventional Schiff-base reaction [51,52]. Through alteration of the sulfonated building block with benzene-1,4-diamine (Pa), 2,5-diaminobenzenesulfonic acid (Pa-SO_3_H) or 4,4′-diaminobiphenyl-3,3′-disulfonic acid (BD-(SO_3_H)_2_), the counterparts of TpPa COF, SCOF-1 and SCOF-3 were synthesized for comparison. Moreover, by enlarging the building block size of Tp with 1,3,5-tris(4-formylphenyl)benzene (TFPB), SCOF-4 was also obtained from the condensation reaction of TFPB and BD-(SO_3_H)_2_ (the detailed synthesis procedure of TpPa COF, SCOF-1, SCOF-3 and SCOF-4 are shown in Appendix A). The crystalline structure of TpPa COF and SCOFs was first probed using the powder X-ray diffraction (PXRD) technique and theoretical calculation structural simulations. The experimental PXRD results demonstrated that SCOF-2 had two well-resolved peaks at 4.25° and 26.23°, corresponding to the (100) and (001) facets, which matched well with the eclipsed AA stacking mode instead of the staggered AB stacking mode. After Pawley refinement, a set of peaks could duplicate the experimental results with satisfied factors of *R*_p_ = 4.450% and *R*_wp_ = 3.742% (Figure 1a and Appendix A). In addition, the experimental PXRD patterns of TpPa COF, SCOF-1, SCOF-3 and SCOF-4 also exhibited characteristic peaks that are highly consistent with the simulated crystallographic structures and the reported works (Appendix A) [53,54,55,56]. These results proved that the TpPa COF and SCOFs were successfully synthesized.

Furthermore, the N_2_ adsorption/desorption isotherms demonstrated that the TpPa COF and SCOF-1 belonged to both type I and IV sorption isotherm profiles, verifying the coexistence of micropores and mesopores structures. By contrast, SCOF-2 displayed a typical type IV isotherm, indicating the presence of mesopore structures (Figure 1b). The Brunauer–Emmett–Teller (BET) surface area of non-sulfonate TpPa COF and SCOF-1 was calculated to be 532 and 274 m^2^ g^−1^ accompanied with an average pore size of 1.9 and 1.1 nm (Appendix A), while SCOF-2 showed a much smaller surface area of 57 m^2^ g^−1^ and the pore size was calculated to be 1.4 nm, which might be attributed to the plentiful sulfonate groups in the pore channel (Appendix A). Additionally, Fourier Transform infrared spectroscopy (FT-IR) was collected to understand their chemical structures. The characteristic peaks at 1582 and 1248 cm^−1^ were observed for the three COFs, which were assigned to the C=C and C-N stretching vibrations, respectively (Figure 1c). As for SCOF-1 and SCOF-2, the newly formed peaks at 1080 and 1025 cm^−1^ were ascribed to the stretching band of O=S=O, demonstrating the existence of sulfonic groups. As shown in the X-ray photoelectron spectroscopy (XPS) of SCOF-2, the peaks of 531.5 eV (O 1s), 400.0 eV (N 1s), 284.7 eV (C 1s), and 167.9 eV (S 2p) were detected (Appendix A).

Moreover, thermogravimetric analysis (TGA) exhibited that TpPa COF, SCOF-1 and SCOF-2 displayed favorable thermal stability under the N_2_ atmosphere, and the weight was retained at 34–49% even up to 800 °C (Figure 1d). Further, the static water contact angles of the synthesized TpPa COF and SCOFs are shown in Figure 1e. Compared with TpPa COF, the water contact angle of SCOF-2 decreased by about 47° after introducing hydrophilic sulfonate groups, which laid the foundation for the selective adsorption of glycopeptides. The zeta potential measurements illustrated the surface charges of the TpPa COF and SCOFs in a buffered solution (pH = 6, the pH value of the loading buffer). The zeta potential of the SCOFs presented negative values of −25.37, −26.13, −16.07 and −19.33 mV, respectively, indicating the strong electronegativity nature of the SCOFs that arise from the sulfonate groups (Appendix A). According to the scanning electron microscopy (SEM) as well as the transmission electron microscopy (TEM) images, the morphology of TpPa COF was composed of short nanofibers (Appendix A); SCOF-2, SCOF-3 and SCOF-4 displayed uniform sheets morphologies (Appendix A); while SCOF-1 showed relatively smooth and long nanofiber morphology (Appendix A). Figure 1f exhibits the high-angle annular dark-field scanning TEM (HAADF-STEM) image and elemental mapping images of SCOF-2, and the components C, N, O, and S were detected concurrently. The TpPa COF and other SCOFs were also characterized by elemental mapping in Appendix A.

### 2.2. Rational Selection of Sulfonate-Rich COFs and TpPa COF for Glycopeptide Enrichment

Compared with the reported traditional enrichment materials, COFs could be precisely customized and preliminarily designed at the molecular level via the rational selection of porous organic building blocks. It could provide theoretical guidance information to screen possible enrichment mediums with hydrophilicity, affinity sites, and molecular linkage within them, which are three probable key factors affecting the enrichment material toward glycopeptides. Therefore, one non-sulfonate TpPa COF and four sulfonate-rich COFs (SCOF-1, SCOF-2, SCOF-3, and SCOF-4) with different topologies and different building blocks were selected as enrichment materials to investigate the selectivity for glycopeptides.

We rationalized the five COFs with hydrophilicity through the water contact angle as the main factor. With the weakest hydrophilicity, TpPa COF showed a water contact angle of 88.7° and could only enrich 10 glycopeptides (Figure 2a). After introducing a hydrophilic sulfonate group to the building block of SCOF-1, the water contact angle decreased by over 55° demonstrating enhanced hydrophilicity. As shown in Figure 2b, 22 glycopeptides with a relatively clean MS background were observed. Moreover, we introduced two functional hydrophilic sulfonate groups to synthesize SCOF-2. Surprisingly, 28 glycopeptides with high signal intensities were detected even though the water contact angle of SCOF-2 was a little higher than SCOF-1, which was probably attributed to the abundant affinity sites between the glycopeptides and the sulfonate group (Figure 2c). Additionally, in order to further evaluate the interactions between the glycan moieties of the glycopeptides and the sulfonate group, we enlarged the size of the building blocks and obtained SCOF-3 and SCOF-4. As depicted in Figure 2d, 21 glycopeptides and a few non-glycopeptides were observed in the MS spectrum, which exhibited a better enrichment selectivity than non-sulfonate TpPa COF but worse than SCOF-2. Amazingly, it is worth noting that SCOF-4 showed the strongest hydrophilicity with a water contact angle of 31.0^o^ among all the enrichment materials. However, only four glycopeptides with low signal intensities were detected, suggesting that SCOF-4 had the worst enrichment selectivity toward glycopeptides (Figure 2e). On the basis of the above results, SCOF-2 was further explored as the best enrichment material for excellent enrichment performance even though its hydrophilicity was not the strongest compared to other SCOF materials.

### 2.3. Theoretical Calculations of Adsorption Modes Between Different COF Models and Monosaccharides

Glycans are diverse structures that are composed of various monosaccharide building blocks including mannose (Man), galactose (Gal), fucose (Fuc), *N*-acetyl-glucosamine (GlcNAc), and others. The primary terminal Neu5Ac unit is one of the most common forms of sialylated glycans, which could interact with the glycans on the proteins to provide adhesion and recognition.

Afterward, the possible adsorption modes between the five COFs and the representative Neu5Ac (Appendix A) were calculated. The adsorption dynamics of Neu5Ac with different COF models were carried out in the CP2K code with density functional theory (DFT) calculation methods [57] (the detailed DFT calculations are shown in Appendix A). As depicted in Figure 3a, one kind of hydrogen bond was formed between the carbonyl moiety in the TpPa COF framework and the hydroxyl of Neu5Ac. After introducing mono-sulfonate in the SCOF-1 framework, three kinds of hydrogen bonds were formed. One was formed between the sulfonate moiety and the hydroxyl of Neu5Ac, the others originated from the interaction between the sulfonate moiety and the carboxylic acid of Neu5Ac (Figure 3b). Noticeably, due to the abundant affinity sites of sulfonate in SCOF-2, four kinds of hydrogen bonds were formed not only between the sulfonate moiety and the carboxylic acid of Neu5Ac but also between the sulfonate moiety and the dihydroxyl of Neu5Ac (Figure 3c). As for SCOF-3, because of the enlarged building block size, the adsorption structure was not stable though the adsorption mode was similar to SCOF-1 (Figure 3d). As depicted in Figure 3e, due to having the largest building block size, only one kind of hydrogen bond was formed between the sulfonate moiety and the carboxylic acid of Neu5Ac, demonstrating that the adsorption mode of SCOF-4 was not stable compared to the other COFs. The side view of the possible adsorption modes between the five COFs and the Neu5Ac is also shown in Appendix A.

Also, the adsorption energy (ΔE) between the Neu5Ac and the different COF models were calculated and the results are shown in Figure 3f. The maximal ΔE of SCOF-2 with -106.02 kcal mol^−1^ corresponded to a stable adsorption structure and the more negative energy value was more conducive to adsorption, which exhibited the highest among the other COF models. The results showed that SCOF-2 exhibited satisfactory chemoselectivity toward Neu5Ac, suggesting the potential enrichment selectivity toward glycopeptides.

### 2.4. Investigation of the Selectivity of SCOF-2 for Glycopeptide Enrichment

In order to evaluate the enrichment performance of SCOF-2 for glycopeptides, a standard model glycoprotein horseradish peroxidase (HRP) tryptic digest was pretreated with SCOF-2 for sequential loading, washing, eluting and finally the eluant was analyzed using matrix assisted laser desorption ionization time of flight mass spectrometry (MALDI-TOF MS). Based on the retention mechanism, the composition proportion of the loading buffer would have a greater influence on the result of glycopeptide enrichment because of the change of polarity. In this work, different compositions of loading buffer were achieved by altering the concentration of ACN (80%, 85%, 90%, and 95%) and TFA (0.05%, 0.1%, 0.5%, and 1.0%) as the mixed solutions were examined (Appendix A). With the proportion of 90% ACN and 0.1% TFA, the number of observed glycopeptides with high signal intensities was gradually obtained, demonstrating that the interactions between SCOF-2 and the glycopeptides became stronger. Consequently, we adopted 90% ACN/0.1% TFA (*v*/*v*) as the optimal loading buffer. Moreover, to obtain the maximum number of glycopeptide enrichment levels using SCOF-2, four different elution buffers (30% ACN and 0.05%, 0.1%, 0.5%, and 1.0% TFA) were investigated using the optimal loading buffer (Appendix A). The greatest number of glycopeptides was obtained when using 30% ACN/0.1% TFA (*v*/*v*) as the elution buffer. Additionally, human serum immunoglobulin G (IgG) was chosen as another model glycoprotein to verify the optimal loading buffer and elution buffer. As shown in Appendix A, considering the number of detected glycopeptides and the signal intensities of glycopeptides, we finally selected 90% ACN/1.5% TFA (*v*/*v*) and 30% ACN/0.5% TFA (*v*/*v*) as the best loading and elution buffer, respectively. The detailed information on the detected glycopeptides from HRP and IgG tryptic digests are summarized in Appendix A, respectively.

Under the optimal enrichment conditions, HRP tryptic digest was first employed for glycopeptide enrichment using SCOF-2. Before enrichment, only five glycopeptide peaks were identified with low signal intensities due to the interference of non-glycopeptides, while 28 glycopeptides appeared with a transparent spectrum background after treatment with SCOF-2 (Appendix A). Furthermore, human serum IgG tryptic digest was also chosen to evaluate the universality of SCOF-2 in glycopeptide enrichment. Similarly, 16 glycopeptides could be observed and the peak intensities of glycopeptides efficiently enhanced after enrichment, while only four peaks of glycopeptides with lower signal-to-noise (S/N) ratios could be detected without enrichment (Appendix A).

### 2.5. Analytical Performance of SCOF-2 for Glycopeptide Enrichment

To further confirm the enrichment performance of SCOF-2, the selectivity and sensitivity of SCOF-2 toward glycopeptides as two vital indicators were also investigated. Bovine serum albumin (BSA) was employed as the interfering non-glycoprotein. Complex samples consisting of HRP tryptic digest and BSA tryptic digest with different molar ratios (1:10, 1:100, 1:1000, and 1:5000) were used to assess its selectivity. As illustrated in Appendix A, when the molar ratios of HRP:BSA = 1:10 and 1:100, 20 and 16 glycopeptides could be detected, though there were a few non-glycopeptides in the spectrum. Remarkably, when the molar ratio was increased to 1:1000, the 11 glycopeptides remained detectable (Figure 4a). It is worth pointing out that even though the molar ratio significantly increased to 1:5000, eight glycopeptides could still be distinctly identified, proving the excellent enrichment selectivity of SCOF-2 toward glycopeptides (Figure 4b). Subsequently, the detection sensitivity of SCOF-2 for glycopeptides was further determined by regularly reducing the concentration of the HRP tryptic digest. When the HRP digest concentration was decreased to 10 fmol μL^−1^ or even at a low concentration of 1.0 fmol μL^−1^, SCOF-2 could still easily enrich the glycopeptides (Appendix A). Then, when the HRP digest concentration was reduced to 0.1 fmol μL^−1^, six glycopeptides remained visible after enrichment (Figure 4c). Moreover, at an even lower concentration of 0.01 fmol μL^−1^, two glycopeptides exhibited predominance in the spectrum (Figure 4d). The results demonstrated that SCOF-2 had exceptional detection sensitivity for glycopeptide enrichment. To sum up, SCOF-2 showed outstanding selectivity and sensitivity towards glycopeptides, better than that of the previously reported hydrophilic materials listed in Appendix A. Thus, it could be permitted for the detection of glycopeptides in genuine complex samples.

The binding capacity for glycopeptides is measured using the reported method [58]. It is one of the key factors for any novel enrichment material, which was investigated by the addition of different amounts of SCOF-2 to a fixed amount of 150 μg HRP digest. The S/N ratios of four selected glycopeptides (*m*/*z* = 2850.7, 3572.8, 3672.1 and 4984.3) in the elution were analyzed using MALDI-TOF MS. It was observed that the S/N ratios of these peaks progressively increased and then maintained saturation with an increasing content of SCOF-2 (Appendix A). Thus, the binding capacity was assumed to be 150 mg g^−1^. This result indicated that SCOF-2 had a high binding capacity for glycopeptides, which might be attributed to its abundant binding sites and outstanding hydrophilicity.

In addition, the reusability and stability of using SCOF-2 for the enrichment of glycopeptides were explored using an HRP tryptic digest. In order to investigate reusability, the previously used SCOF-2 was rinsed with an elution buffer to remove residues before each enrichment step. As depicted in Appendix A, compared with the first cycle, minimal changes were observed of the obtained glycopeptide numbers and signal intensities in the spectrum after five cycles. Even after being stored for two weeks at room temperature, SCOF-2 exhibited the same excellent enrichment performance as the first time. A typical glycopeptide was selected as an indicator and recorded the signal intensities, it could be clearly noticed that the intensities of the glycopeptide peaks changed slightly (Appendix A), which indicated its great reusability and long-term stability. The quantitative stable isotope dimethyl labeling method was investigated to estimate the recovery. As shown in Appendix A, the enrichment recovery (L/H) was determined using the peak intensity of the light-tagged glycopeptide with its heavy-tagged counterpart and the recovery of SCOF-2 for glycopeptides was measured at 89.1%, demonstrating the excellent recovery capability of SCOF-2. The result proved the great potential of SCOF-2 for the analysis of glycopeptides in complex real biological samples.

### 2.6. Application of SCOF-2 in Glycopeptide Enrichment from Tryptic Digests of Proteins Extracted from Human Serum

Human serum is easily obtainable and appropriate for clinical testing, thus the isolation and subsequent identification of glycopeptides in serum can be utilized for the discovery of tumor biomarkers, which can provide new approaches for developing diagnostic and therapeutic strategies. Here, we analyzed glycopeptides in human serum samples from ovarian cancer patients (Group HK, *n* = 3) and healthy volunteers (Group CK, *n* = 3) and performed three parallel experiments using the SCOF-2 workflow. After capturing glycopeptides in the serum of ovarian cancer patients and healthy volunteers with SCOF-2, the obtained peptides were analyzed with nano liquid chromatography mass spectrometry/mass spectrometry (nanoLC-MS/MS). The common numbers of 196 glycopeptides and 227 glycosylation sites mapping to 82 glycoproteins were identified in the ovarian cancer patients, compared with 194 glycopeptides and 225 glycosylation sites mapping to 82 glycoproteins in the healthy volunteers (Figure 5a and Appendix A, the details were listed in Appendix A). The Venn diagram in Appendix A summarizes the glycopeptides, glycoproteins and glycosylation site enrichment performance of SCOF-2 in three experimental replicates of human serum from ovarian cancer patients and healthy volunteers. Noticeably, it is worth noting that over 80% of the glycopeptides from ovarian cancer patients and healthy volunteers were mono-glycosylation sites while multi-glycosylation events per peptide were identified in less than 4% of the glycopeptides (Appendix A).

In order to understand the biological functions of various genomes and evaluate the biological significance of the identified glycoproteins, we investigated the gene ontology (GO) enrichment by using the gene ontology database (Appendix A). In biological processes, glycoproteins involved in innate immune response were up-regulated in ovarian cancer patients compared to healthy controls, revealing differences between cancer patients and healthy individuals. On the contrary, in molecular functions, glycoproteins involved in immunoglobulin receptor binding were down-regulated in ovarian cancer patients compared to healthy controls. In terms of cellular components, glycoproteins were mostly correlated to peptides in the blood microparticle GO term, which corresponded to the fact that the HK and CK sample groups were derived from human serum.

To comprehensively evaluate the differences between ovarian cancer patients and healthy controls, a versatile statistical analysis and quantitative comparison of the expression level of the protein PTMs were explored. Principal component analysis (PCA) based on the enrichment results of quantitative comparison proved that the HK and CK groups were partly separated, revealing the noteworthy differences between the ovarian cancer patient group and the healthy control group. The results of the three parallel experiments of the healthy control group were comparatively discrete, indicating more important individual differences in the healthy control patients (Figure 5b). In agreement with PCA, ovarian cancer patients and healthy volunteers showed heterogeneous mapping of glycoprotein abundance (Figure 5c). The expression levels of the protein outlines of ovarian cancer patients and healthy controls captured by SCOF-2 form distinct clusters when employing hierarchical cluster analysis (HCA), with the clusters colored in blue, red, and orange. In brief, the excellent experimental results mentioned above demonstrate a unique distribution profile of glycopeptides enriched by SCOF-2, which could find cancer-specific relationships between healthy controls and ovarian cancer patients, allowing the identification of target glycoproteins.

## 3. Discussion

Aberrant protein glycosylation is closely associated with a number of biological processes and diseases. However, characterizing the types of post-translational modifications (PTMs) from the complex biological samples is challenging for comprehensive glycoproteomic analysis [59]. Therefore, the selective capture of low-concentration glycoproteins and glycopeptides from complex mixtures is a significant tool for in-depth glycoproteome researchers. To date, plenty of HILIC-based materials, such as graphene oxide [60], polymer nanoparticles [61], and metal–organic frameworks [62] have been widely reported. Unfortunately, due to the limited glycopeptide-specific recognition sites, unsuitable mass transfer kinetics and low relative density of hydrophilic groups, traditional HILIC materials usually suffer from low glycopeptide binding selectivity and low detection sensitivity.

Covalent organic frameworks (COFs), as a class of long-range ordered porous organic materials, have shown great potential in many aspects owing to the extensive tunability [63]. The sulfonyl group exhibited excellent hydrophilicity and chemical stability; however, sulfonyl-functionalized COFs for the selective enrichment of glycopeptides have rarely been reported. Therefore, in this work, we attempted to functionalize the as-prepared COFs. Herein, sulfonate-rich COFs and a non-sulfonate TpPa COF were introduced. SCOF-2 together with another three, SCOF-1, SCOF-3, and SCOF-4, was obtained for the successful enrichment of glycopeptides.

Our study found that a total of 28 and 16 glycopeptides could be efficiently detected from HRP and IgG tryptic digest, respectively. Moreover, the results of the theoretical calculations were consistent before the experiment. The as-prepared SCOF-2 has an ultralow detection limit (0.01 fmol μL^−1^), excellent enrichment selectivity (molar ratio HRP:BSA = 1:5000), satisfactory recovery rate (89.1%), high adsorption capacity (150 mg g^−1^) and good reusability in the individual enrichment. Meanwhile, by using SCOF-2 adsorbent, 196 and 194 endogenous glycopeptides in the serum of ovarian cancer patients and healthy people were successfully enriched and identified. The incorporation of multiple hydrophilic sulfonate groups within the SCOF-2 structure induces a substantial enhancement in surface hydrophilicity. This hydrophilicity amplification originates from the synergistic effects of the formation of hydrogen-bonding networks through exposed sulfonic acid moieties. Notably, the high areal density of these hydrophilic functionalities creates a spatially ordered recognition matrix that exhibits exceptional glycopeptides affinity. These findings position SCOF-2 as an outstanding material for phosphoproteomic studies, particularly in low-abundance glycopeptides detection, which exhibits excellent enrichment performance than the existing methods and other enrichment strategies.

In summary, we constructed a novel hydrophilic sulfonate-rich COF (SCOF-2) for the enrichment of glycopeptides, demonstrating the enrichment performance from standard protein digests with good specificity, high sensitivity and outstanding stability. It also demonstrated excellent enrichment capacity and reproducibility toward glycopeptides. As a result, SCOF-2 could successfully enrich 196 glycopeptides from the human serum of ovarian cancer patients, revealing its superiority and feasibility in the selective enrichment of glycopeptides. Amazingly, proteomic analysis of the captured proteins proved that it was possible to distinguish healthy controls from ovarian cancer patients. In addition, the excellent performance of SCOF-2 in the application of complex biological samples could provide great potential for the early clinical diagnosis of disease biomarkers caused by abnormal protein glycosylation.

## 4. Materials and Methods

### 4.1. Synthesis of SCOF-2

SCOF-2 was prepared according to previously reported works procedure with a slight modification [51,52]. Typically, 63 mg (0.3 mmol) of triformylphloroglucinol (Tp) and 120.6 mg (0.45 mmol) of 2,5-diaminobenzene-1,4-disulfonic acid (Pa-(SO_3_H)_2_) were added to a Pyrex tube in presence of 1.5 mL 1,4-dioxane, 1.5 mL mesitylene and 0.5 mL of 6 M aqueous acetic acid (AcOH). This mixture was then sonicated for 20 min to form a homogeneous suspension. The tube was subsequently flash-frozen under liquid nitrogen temperature (77 K) and degassed by three freeze–pump–thaw cycles. Then, the reaction mixture was sealed and heated at 120 °C for 72 h under a static condition. After cooling, the deep brown precipitate was collected by filtration and washed with copious amounts of dimethylacetamide, deionized water and anhydrous tetrahydrofuran. The material was then dried under vacuum at 120 °C for 12 h to obtain SCOF-2 as a deep brown powder. The detailed synthesis procedure of TpPa COF, SCOF-1, SCOF-3 and SCOF-4 are shown in Appendix A.

### 4.2. Enrichment of Glycopeptides from Tryptic Digests of Standard Proteins

The detailed process of glycopeptides enrichment was described as follows. Firstly, 1.0 mg SCOF-2 was placed in a centrifuge tube and ultrasonically dispersed in 100 μL of loading buffer (90% ACN/0.1% TFA (*v*/*v*) for HRP tryptic digest or 90% ACN/1.5% TFA (*v*/*v*) for IgG tryptic digest). Then, 1.0 μL tryptic digest of standard protein was added to the centrifuge tube and the mixture was incubated at 37 °C for 30 min. After enrichment, SCOF-2 was then separated by centrifugation and washed three times with loading buffer to remove non-glycopeptides. Finally, 10 μL elution buffer (30% ACN/0.1% TFA (*v*/*v*) for HRP tryptic digest or 30% ACN/0.5% TFA (*v*/*v*) for IgG tryptic digest) was added to collect the adsorbed hydrophilic glycopeptides and analyzed using MALDI-TOF MS.

### 4.3. Enrichment of Glycopeptides from Tryptic Digest of Human Serum

Enrichment of glycopeptides from tryptic digests of human serum samples was similar to the above procedure. In the first place, lyophilized human serum digest was redissolved in 100 μL of loading buffer (90% ACN/0.1% TFA (*v*/*v*)) and incubated with 1.0 mg of SCOF-2 for 30 min. Then, the mixture was centrifugated for 6 min to remove supernatant and washed with 100 μL of loading buffer three times to remove non-glycopeptides. Thereafter, the captured glycopeptides were eluted using 10 μL elution buffer (30% ACN/0.1% TFA (*v*/*v*)) for 10 min. Finally, the collected solution was lyophilized and desalted for further deglycosylation and nanoLC-MS/MS analysis.

### 4.4. Contact Angle Measurement

The process involves the following steps: first, the compounds are milled and then it is compacted and uniformly deposited onto the substrate. The water drop with a defined volume is placed on the compounds, and then the photograph is taken and the angle between the tangent to the water drop and the substrate is measured. The angle can be measured manually with the software.

### 4.5. MALDI-TOF MS Analysis

All MALDI-TOF MS analyses were performed on a Bruker autoflex speed time-of-flight mass spectrometer in a positive reflection mode with an Nd% YAG laser at 355 nm, a repetition rate of 200 Hz and an acceleration voltage of 20 kV in the *m*/*z* range of 2000–5200 and analyzed using flexAnalysis software (version 3.3). A volume of 1 μL eluent and 1 μL matrix solution (α-cyano-4-hydroxycinnamic, CHCA, 10 mg mL^−1^) were mixed and deposited on an AnchorChip standard MALDI plate for MALDI-TOF MS analysis. The detailed methods of LC–MS/MS are shown in Appendix A.

## Data Availability

The data that support the findings of this study are available from the corresponding authors upon reasonable request.

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
