# Peer review of "Hydrophilic Sulfonate Covalent Organic Frameworks for Serum Glycopeptide Profiling"

_ijms, 2025, doi:10.3390/ijms26051957_

Round 1
Reviewer 1 Report
Comments and Suggestions for Authors
The study presents a novel sulfonate-rich covalent organic framework (COF) named SCOF-2 for enhancing the identification of glycopeptides in complex biological mixtures. Aberrant glycosylation is highlighted as a significant indicator of various diseases, necessitating effective profiling techniques. SCOF-2 was synthesized using a Schiff base reaction and demonstrated remarkable properties, including high hydrophilicity, low detection limits, excellent selectivity, and reusability in enrichment processes. The material successfully identified 196 glycopeptides in ovarian cancer patients' serum and 194 in healthy individuals, showcasing its potential application in glycoproteomics. The findings suggest that SCOF-2 may help unravel the glycosylation landscape in biological research, enhancing mass spectrometry's effectiveness for glycopeptide analysis. However, by analyzing the content of this study, here are some potential errors or areas of concern that may need to be addressed:
- The paper has some inconsistencies in terminology and syntax that could confuse readers. For example, phrases such as "the first attempt" may sound vague without context.
- While the synthesis of SCOF-2 is mentioned, the details could be more precise. Specific temperature control, duration, and reagents must be carefully outlined to replicate the study.
- Phrases such as "the theoretical calculations were consistent with experimental" should include "the" before "experimental": "theoretical calculations were consistent with the experimental”.
- In various sections, phrases are repeated unnecessarily, such as "it was first introduced here in this work", which could be simplified.
- Sentences that start with "However, characterization the type of post-translational modifications" are grammatically incorrect and should be rephrased (e.g., "characterizing the types of post-translational modifications is challenging”).
- The handling of lists, especially when detailing multiple points (like in methods or results), often lacks clarity due to missing commas.
These observations indicate that while the foundational scientific work appears thorough, the language and typographical elements should be improved for clarity and professionalism. Adjusting these aspects would greatly help enhance the document's overall readability.
Comments on the Quality of English Language
- The paper has some inconsistencies in terminology and syntax that could confuse readers. For example, phrases such as "the first attempt" may sound vague without context.
- Phrases such as "the theoretical calculations were consistent with experimental" should include "the" before "experimental": "theoretical calculations were consistent with the experimental”.
- In various sections, phrases are repeated unnecessarily, such as "it was first introduced here in this work", which could be simplified.
- Sentences that start with "However, characterization the type of post-translational modifications" are grammatically incorrect and should be rephrased (e.g., "characterizing the types of post-translational modifications is challenging”).
- The handling of lists, especially when detailing multiple points (like in methods or results), often lacks clarity due to missing commas.
These observations indicate that while the foundational scientific work appears thorough, the language and typographical elements should be improved for clarity and professionalism. Adjusting these aspects would greatly help enhance the document's overall readability.
Reviewer 2 Report
Comments and Suggestions for Authors
The manuscript ijms-3439244 investigated hydrophilic sulfonate covalent-organic frameworks for serum glycopeptide profiling. The study of hydrophilic sulfonate covalent organic frameworks to detect glycopeptides has been previously described:
Yin Ji, Heming Li, Jinghan Dong, Jiashi Lin, Zian Lin, Super-hydrophilic sulfonate-modified covalent organic framework nanosheets for efficient separation and enrichment of glycopeptides, Journal of Chromatography A, Volume 1699, 2023, 464020, ISSN 0021-9673, https://doi.org/10.1016/j.chroma.2023.464020.
On this matter, the present manuscript does not offer innovative knowledge of hydrophilic sulfonate covalent organic frameworks.
Reviewer 3 Report
Comments and Suggestions for Authors
The manuscript investigates the use of sulfonate-rich covalent organic frameworks (SCOF-2) for the enrichment of glycopeptides, particularly in the context of ovarian cancer biomarker discovery. The results demonstrate that SCOF-2 enhances glycopeptide enrichment efficiency, sensitivity, and selectivity.
This study presents a novel approach to glycopeptide enrichment by introducing sulfonate-functionalized COFs. The application to ovarian cancer biomarker discovery adds clinical relevance.
Despite the importance of the work, some details deserve attention.
- Clarity and Justification: While the rationale for using SCOF-2 is compelling, a clearer explanation of why this approach surpasses existing methods would strengthen the introduction. Introduction missing background on alternative glycopeptide enrichment methods.
- Methodological Robustness: The manuscript lacks some experimental details necessary for full reproducibility, such as precise parameters for mass spectrometry settings. Some procedural details are missing, such as specific incubation times and buffer compositions.
- Data Interpretation: Experimental limitations include a lack of statistical comparisons between SCOF-2 and other enrichment materials. The discussion could be expanded by comparing SCOF-2 performance with other enrichment strategies in terms of statistical significance.
Reviewer 4 Report
Comments and Suggestions for Authors
The document titled “Hydrophilic Sulfonate Covalent-Organic Frameworks for Serum Glycopeptide Profiling” presents significant research and results above the possible markers to detect patients with cancer trough a simple test. The document is adequate for this journal, below are listed some recommendations in order to improve the document and in general the authors should consider to incorporate some results from supporting information in the main text since these are of huge relevance to have better scope of the research.
- please explain or improve the figure 1a since this is not completely clear as well as the structure of each compounds
-the line 112 maybe does not have the complete text
-Please specify in the main text that procedure of synthesis which is include in supporting information
-Do you have the results of X-rays of monocrystal? if you have these please include them
-in line 150-151, you mentioned that compounds present a good thermal stability at 800 C but, are they the intial chemical structures or are a residue at this temperature?
-How was measured the static contact angles of the compounds? These are powders, please include the methodology used
-explain how the glycopeptides were calculated for each structure
-please define all abbreviations
- Please cite supporting information for DTF details
-in section 2.4 or before section, explain that it will be used de SCPOF-2 for subsequent studies due to this present the better results
-improve the figure 5, the tags are difficult to understand due to the size
Reviewer 5 Report
Comments and Suggestions for Authors
This article describes the synthesis and characterization of novel covalent sulfonate-rich organic frameworks (SCOF) for the enrichment of glycopeptides. The authors describe the synthesis, molecular characterization of the structure of the SCOFs, including DFT calculations, and then compare the SCOF ability to enrich glycopeptides from model peptide digests. Finally, the authors compare the synthesized SCOFs for glycopeptide enrichment from healthy vs cancer samples and conclude that their SCOFs show promising enrichment and are a potential good candidate for further investigation.
The article is of decent scientific presentation, though I have some difficulty in several places understanding the intention of the authors - certainly a translation issue that complicates my understanding. Most importantly, there are significant missing details on in the methods section, entire aspects of the described experimnts are missing such as the details on the DFT calculations and the nanoLC-MS/MS. Without these details the manuscript is unacceptable and I recommend rejection.
Minor concerns:
Line 24: HRP needs to be defined in the abstract
Line 91: "Sulfonyl groups exhibit ... glycopeptides have been..." - present tense description.
Line 93: change to "Here in this work... " and revise the sentence
Line 102: "Treated" is the wrong word choice here as it was used to separate the samples not treat a subject
Scheme 1: The icons in the scheme in b are very tiny - difficult to see even at 200% magnification on my screen, could the authors improve?
Scheme 1: the caption requires a more complete description of the scheme
Line 112: "The text continues here???"
Line 124: Well resolved instead of reserved?
Line 137: no need to define the acronym if not used again
Line 151: remove "the temperature"
Line 156: "in deionized water" but at pH 6 - no longer deionized was it buffered?
Lien 186: suggest change "and significantly" to "demonstrating"
Figure 2: why the break bar in the y-axis when the scale looks complete?
Paragraph starting at line 213 needs careful rewriting especially regarding the hydrogen bonds. Are these different kinds of bonds or just numbers? are they predicted or formed?
Line 232: Remove "in this experiment"; also the delta E was calculated and the results are
Line 236: The authors discuss chemoselectivity here but have shown no evidence - are they referring to later results?
Figure 3: It would be nice if the structures were labelled in the figure.
Line 285: The authors discuss the detection limit of their SCOF material - this will be highly method dependent so it is inappropriate term to use here.
Line 302: How was binding capacity calculated - missing from discussion and methods
Line 330: "extensive and sensitive??" I don't understand the intended meaning
Line 343: "technique" should be "technical" though I don't think that is the correct wording - these are biological replicates, unless the MS data are technical replicates as well - if the methods were complete it would give a clue
Line 348: "peptide sequences"
Line 361: "correlated to peptides in the blood microparticle GO term"
Line 366: "protein PTMs"
Line 369: "results of three parallel experiments" do the authors mean the three samples? or three separate experiments?
Figure 5 and text: I do not understand the logo diagram - what is the origin of the sequences for comparison? Also CK1 is missing from (c)?
Line 412: ends in an incomplete sentence
Line 442: what concentration of 1 uL digest? total amount?
Line 443: what temp was the incubation?
Major concerns:
- Figure 4 and relevant text - How were the peaks assigned as glycopeptides?
- Line 337 and methods: the nanoLC-MS details are absent. The manuscript cannot be accepted with such large missing information
- The MALDI experiments are also inadequately described. For example, all the MALDI settings used, the software used to collect, process, and visualize the data.
- Informed consent statement: The authors claim no informed consent was required and also give no details on the source of the ovarian and healthy human samples.
Comments on the Quality of English Language
Needs to be carefully revised - as above and throughout.
Round 2
Reviewer 4 Report
Comments and Suggestions for Authors
The document is suitable for publication, congratulations
Reviewer 5 Report
Comments and Suggestions for Authors
1. Line 349: The authors mention leucine and serine, did they mean threonine (N-X-T)? Regarding the logo diagram - I am still confused how consensus sequences were even created from Table S4. How are the peptides classified as either HK or CK? At least in the case of the ovarian cancer peptides I can see the two specific peptides, but there are many sequences in the logo diagram. Overall, I don't even see that it is adding anything but confusion and suggest removing it from the paper.
2. The authors respond: "CK1 was deleted in Figure 5c due to the low individual differences." Even though I do not fully comprehend what the authors mean by this statement, it is essential this is included in the MS text. My understanding is that the PCA analysis should have all points being compared - is CK1 an outlier or does it exactly overlay another point?
3. A statement in the methods of the manuscript is required to indicate that there are more details on the synthesis and analysis in the SI materials. Specifically, the LC-MS/MS methods, which this reviewer feels should be included in the main methods, but leave it to the authors.
4. "All analyses were performed by a timsTOF HT mass spectrometer (Bruker) equipped with an Easyspray source (Thermo, USA)."
I was under the impression that Thermo did not supply an easyspray source for the Bruker TimsTof - I could be mistaken, please double check this.
Comments on the Quality of English Language
The updates are satisfactory in regards to English - careful editing is still recommended.